# Maternal Antioxidant Status in Early Pregnancy and Development of Fetal Complications in Twin Pregnancies: A Pilot Study

**DOI:** 10.3390/antiox9040269

**Published:** 2020-03-25

**Authors:** David Ramiro-Cortijo, María de la Calle, Pilar Rodríguez-Rodríguez, Ángel L. López de Pablo, María R. López-Giménez, Yolanda Aguilera, María A. Martín-Cabrejas, María del Carmen González, Silvia M. Arribas

**Affiliations:** 1Department of Physiology, Faculty of Medicine, Universidad Autónoma de Madrid; C/ Arzobispo Morcillo 2, 28029 Madrid, Spain; dramiro@bidmc.harvard.edu (D.R.-C.); pilar.rodriguezr@uam.es (P.R.-R.); angel.lopezdepablo@uam.es (Á.L.L.d.P.);; 2Obstetrics and Gynecology Service, La Paz University Hospital; Paseo de la Castellana 261, 28046 Madrid, Spain; Maria.delacalle@uam.es; 3Department of Preventive Medicine, Public Health & Microbiology, Faculty of Medicine, Universidad Autónoma de Madrid; C/ Arzobispo Morcillo 2, 28029 Madrid, Spain; Mrosario.lopez@uam.es; 4Department of Agricultural and Food Chemistry-CIAL, Faculty of Sciences, Universidad Autónoma de Madrid; Ciudad Universitaria de Cantoblanco, 28049 Madrid, Spain; Yolanda.aguilera@uam.es (Y.A.); maría.martin@uam.es (M.A.M.-C.)

**Keywords:** antioxidants, cortisol, FGR, melatonin, prematurity, twin gestations.

## Abstract

Twin pregnancies are increasing due to the rise in mothers’ childbearing age and have a higher risk of fetal growth restriction (FGR) and prematurity. Therefore, early prediction of these events is important. Our aim was to analyze in the first trimester of pregnancy a possible association between antioxidants, including melatonin, in maternal plasma and the development of fetal complications in twin pregnancies. A single-center, prospective, and observational study was performed in 104 twin-pregnant women. A blood sample was extracted between the 9th and the 11th week of gestation, and plasma was obtained. Antioxidants (thiols, reduced glutathione, phenolic compounds, catalase, superoxide dismutase) and oxidative damage biomarkers (carbonyl groups and malondialdehyde) were assessed by spectrophotometry, and global scores were calculated from these parameters (Antiox-S, Prooxy-S). Melatonin and cortisol were evaluated by a competitive immunoassay. In the first trimester of pregnancy, Antiox-S was significantly lower in women who developed FGR compared to those with normal fetal growth; plasma melatonin was significantly lower in women with preterm compared to those with full-term births and exhibited a positive correlation with birth weight. Maternal cortisol showed a negative correlation with birth weight. We conclude that, for twin gestations, maternal plasma antioxidant status and melatonin could be potential biomarkers to be included in algorithms to predict FGR and preterm labor.

## 1. Introduction

The number of twin births has increased over the last decades in many industrialized countries [1], linked with older childbearing age, decline in fertility, and subsequent use of assisted reproduction techniques (ART) [2,3]. Twin pregnancies have higher incidence of fetal growth restriction (FGR) and prematurity [4,5,6], which increases the risk of perinatal morbidities and long-term development of cardiometabolic diseases [7,8]. Therefore, finding parameters that enable the prediction of fetal complications associated with twin pregnancies is an important issue in obstetric research [1].

The majority of the studies evaluating potential predictive biomarkers of maternofetal complications have been performed in single gestations, being twin pregnancies usually excluded because they are a source of confounding factors. Therefore, our aim was to explore maternal plasma in early pregnancy in search of potential biomarkers related to oxidative status, which could be associated with the development of fetal complications in twin pregnancies. We focused on maternal oxidative status, since the maintenance of an adequate balance between reactive oxygen species (ROS) and antioxidants is critical in pregnancy [9], and oxidative stress is linked to maternofetal complications and deficient fetal growth [10,11]. We have previous evidence that a poor antioxidant status in early pregnancy is associated with the development of maternal complications, including preeclampsia, gestational hypertension, and gestational diabetes [12]. Moreover, melatonin is a relevant antioxidant hormone directly acting as a ROS scavenger, also able to increase the expression of several antioxidant enzymes [13], and with important roles in pregnancy [14,15].

Glucocorticoids administered before birth have beneficial effects on fetal organ maturation. However, in excess, they also have the potential to reduce fetal growth and program cardiometabolic diseases, as demonstrated by numerous animal studies [16]. Cortisol access to the fetus is limited by the placental barrier enzyme 11-β hydroxysteroid dehydrogenase 2 (11-β-HSD2), which converts biologically active cortisol into inactive cortisone. The activity of this enzymatic system may be reduced in adverse conditions such as maternal stress, hypoxia, and nutritional restriction [17], and impairment of the placental glucocorticoid barrier has been associated with FGR [18].

In this pilot study, we aimed to explore the possible association between plasma antioxidants (including melatonin) and cortisol, and the later development of FGR or premature labor in healthy women with twin pregnancies, in the first trimester of gestation. For those variables which exhibited significant differences between groups, we also evaluated possible reference points through Receiver Operating Curves (ROC), which could help to evaluate the potential use of a variable as a predictive parameter. This investigation could help to establish potential biomarkers to include in algorithms for early prediction of fetal complications in twin pregnancies.

## 2. Materials and Methods

### 2.1. Population of Study

This was a single-center, prospective, observational pilot study, including 104 healthy twin-pregnant women who attended the Obstetrics and Gynecology Service of La Paz University Hospital (HULP, Madrid, Spain). The study was performed in accordance with the Declaration of Helsinki regarding studies in human subjects and it was approved by HULP (Madrid, Spain; HULP: PI-1490) and Ethical Committees of the University Autónoma de Madrid (Spain; CEMU, 2013-10).

Recruitment was performed in primiparous twin-pregnant women at the 9th week of gestation. Inclusion criteria were healthy women with twin pregnancy (monochorionic and dichorionic), no previous history of obstetric or fetal complications, normal body mass index (<24.9). Women who agreed to be part of the study signed an informed consent, where the aims of the study and the confidentiality of the data were explained. In the first interview, the women answered a personal questionnaire about maternal age, nationality, employment status, and level of education, and were scheduled for a blood sample extraction between the 9th and the 11st week of gestation, matching it with a routine analysis. The women were followed up at the High-Risk Pregnancy Unit of HULP until delivery, recording the following data: use of ART, gestational age, development of fetal complications (FGR and preterm labor), and birth weight. FGR was defined as fetal growth <3rd percentile or <10th percentile, with hemodynamic alterations assessed by Echo-Doppler. This evaluation included fetal biometric parameters, and umbilical artery, middle cerebral artery, and uterine artery blood flow. Preterm labor was defined as birth at gestational age ≤37 weeks.

Blood samples were extracted between the 9th and the 11st week of gestation, in fasting state, from 8:00 to 9:00 a.m., following the protocols established by the medical staff. The extraction was performed by venipuncture using Vacutainer^®^ tubes containing lithium heparin and a separation gel. The plasma was obtained by centrifugation (2100 rpm, 15 min at 4 ℃), within a maximum of 2 h after extraction, immediately aliquoted, and stored at −80 ℃ until use.

### 2.2. Maternal Blood Parameters

Hematology and biochemistry. Plasma concentrations of glucose (mg/dL), cholesterol (mg/dL), and triglycerides (mg/dL) as well as hematocrit (%) and leucocytes (10^6^/mL) were measured by the Laboratory Medicine Service of HULP.

### 2.3. Maternal Plasma Oxidative Status Parameters

Catalase activity. Catalase activity was assessed by Amplex Red catalase assay (EnzChek Myeloperoxidase Assay Kit with Amplex Ultra Red reagent; Invitrogen, Thermofisher, Madrid, Spain). Catalase activity was expressed as U/mg of protein. Protein content was assessed by a Coomassie-blue-based microtiter plate assay, according to the manufacturer’s instructions (Bio-Rad, Madrid, Spain).

Superoxide anion scavenging activity (SOSA). SOSA assay was used as global measure of superoxide dismutase (SOD) activity. SOSA was determined using a luminescence assay with coelenterazine as a detection probe, adapted to a microplate reader (Synergy HT Multimode; BioTek, Potton, UK). SOSA values were quantified by comparing the luminescence inhibition of each sample with SOD activity standard curve (0–4 U/mL) and expressed as mU SOD/mg of protein.

Plasma reduced glutathione (GSH). Plasma GSH was assessed by a fluorimetric micro-method based on the reaction with o-phthalaldehyde. Fluorescence was measured in the microplate reader at 360 ± 40 nm excitation and 460 ± 40 nm emission wavelengths. GSH concentration of the samples was expressed as μmol/mg of protein.

Total thiol groups. Plasma thiols were assessed by a 5,5′-dithiobis(2-nitrobenzoic acid) assay, adapted to a microplate reader. The absorbance was measured at 412 nm in the microplate reader, and thiol content was expressed as nM GSH/mg of protein.

Phenolic compounds. The Folin–Ciocalteu assay, modified to remove protein interference, was used to assess food-derived antioxidants, mainly phenolic compounds and ascorbic acid. Absorbance was measured at 760 nm, and the results were expressed as mg gallic acid equivalents (GAE) per liter (mg GAE/L).

Plasma malondialdehyde (MDA). The concentration of plasma MDA was measured by a spectrophotometric method detecting thiobarbituric acid (TBA) reactive substances. The samples were incubated with Trichorloacetic acid (TCA), Ethylenediaminetetraacetic acid (EDTA), Sodium dodecyl sulfate (SDS), and Butylated hydroxytoluene (BHT), followed by addition of TBA and boiled in a water bath at 100 °C for 30 min. After cooling, the mixture was centrifuged at 10,000 g, and the absorbance was measured at 532 nm in the microplate reader and compared with values of the standard curve of 1,1,3,3-tetrathoxypropane. Results were expressed as µmol/L.

Carbonyl groups. Plasma protein carbonyl groups were assessed with a 2,4-dinitrophenylhydrazine-based assay. Protein carbonyl concentration was determined using the extinction coefficient of 2,4-dinitrophenylhydrazine (ε = 22,000 M/cm) and expressed as nmol/mg of protein. The absorbance was measured at 370 nm in the microplate reader.

Assessment of global antioxidant and pro-oxidant status. The scores to assess global antioxidant and pro-oxidant status (Antiox-S, Prooxy-S) were calculated taking into account all the antioxidants (catalase activity, SOSA, thiol groups, GSH, phenolic compounds, and melatonin) or oxidative damage biomarkers (MDA and carbonyl groups) measured in plasma. The calculation uses a statistical methodology previously described, which normalizes and standardizes each of the parameters of interest, enabling to sum parameters of different units [12]. Briefly, each parameter (j) was transformed through the logarithmic function (log j). Thereafter, each parameter was standardized for each subject (i) according to the following equation:Z_ij_ = (X_ij_ − M_j_)/STDV_j_
where Z_ij_ is the standardized parameter, X_ij_ is the raw normalized parameter for each subject, M_j_ and STDV_j_ are the mean and the standard deviation of the population, respectively.

### 2.4. Maternal Plasma Hormones

Cortisol. A competitive immunoassay using direct chemiluminescent technology was used, and analysis was carried out with an Advia Centaur instrument (Siemens Healthineers).

Melatonin. Plasma was evaporated to dryness with an evaporator centrifuge (Speed Vac SC 200; Savant, CT, USA), and the residues were dissolved in distilled water. Thereafter, melatonin concentrations were determined by a competitive enzyme immunoassay kit (Melatonin ELISA; IBL International, Hamburg, Germany), according to the manufacturer’s instructions, and expressed as pg/mL.

### 2.5. Statistical Analysis

This was a pilot observational study, which included pregnant women recruited during a 2-year period. Database and statistical analysis were performed with SPSS software (version 24.0; IBM Company, Armonk, NY, USA). The quantitative variables were expressed as mean ± standard error of mean (SEM) or as median and interquartile range (IQR), according to the distribution of the data, and the comparison between groups were tested by Student t test or Mann–Whitney U test, respectively. Spearman correlations (*r*) were used to analyze the relationship between quantitative variables. The significance level was established at *p* < 0.05. For those variables exhibiting significant differences between groups, we evaluated possible reference points by ROC to evaluate the capacity of the variable as a predictive parameter.

## 3. Results

### 3.1. Population of the Study

In this study, 104 pregnant women with twin pregnancies were included, 91.5% of Caucasian origin and the rest of South American or North African origin. Regarding educational and economic level, 70.0% went to college, 85% were employed, and 74.1% had middle-high income. Table 1 describes the maternal and neonatal characteristics of the population.

### 3.2. Maternal Blood Parameters

We did not detect differences in any of the hematological or biochemical parameters evaluated between women who developed FGR or preterm labor and their specific control groups (normal growth or full-term) (Table 2).

### 3.3. Maternal Oxidative Status Parameters

We did not detect differences in the levels of individual maternal antioxidants between women who developed FGR or preterm labor and their specific control groups (Table 3). However, Antiox-S was significantly lower in women who developed FGR compared to those with normal fetal growth (*p-value* = 0.036), while no differences were observed between women with full-term labor and those with preterm labor (Figure 1A). According to the ROC, with a true-positive rate (sensitivity) of 75.0% and a false-positive rate of 19.4%, the threshold to detect FGR in twin pregnancies for Antiox-S was −0.35 arbitrary units.

No difference in Prooxy-S value was found between women who developed FGR and women with normal fetal growth or between women who had preterm labor versus women with full-term labor (Figure 1B). 

Maternal Antiox-S exhibited a positive correlation with birth weight of the second neonate. No significant correlation was found between Prooxy-S and birth weight of the first or second neonate (Figure 2).

### 3.4. Maternal Hormones

We did not detect a statistical difference in maternal cortisol levels between women who developed FGR or preterm labor and their specific control groups (Figure 3A). No significant differences in melatonin level were detected when comparing women with FGR with those with normal fetal growth. However, maternal melatonin was significantly lower in women who had preterm labor compared with those with full-term labor (Figure 3B). With a true-positive rate (sensitivity) of 71.1% and a false-positive rate of 17.3%, the threshold for melatonin to detect preterm labor in twin pregnancies was 4.61 pg/mL in the ROC.

A significant and positive correlation was found between maternal plasma melatonin and birth weight of both first and second neonate. On the other hand, a significant negative correlation was found between maternal plasma cortisol and birth weight of the second neonate (Figure 4).

## 4. Discussion

This pilot study aimed to evaluate if maternal plasma antioxidant status in early pregnancy is associated with FGR or preterm labor in twin gestations, in order to evaluate its potential to be included in predictive algorithms. This is a relevant population, since it has a high risk of fetal and maternal complications, and the rate of twin gestations is increasing due to the rise in childbearing age and the use of ART. Our main findings are that, in women without prior history of maternofetal complications, low maternal score of antioxidant status (Antiox-S) and plasma melatonin levels in the first trimester of pregnancy are associated with later development of FGR and preterm labor. Therefore, maternal antioxidants could be considered as potential biomarkers.

The average age of the women in our population was 35 years, and over 66% of twin gestations were derived from assisted reproduction. This reflects the sociodemographic pattern of western societies, particularly in urban areas, where there is a high childbearing age—mainly due to work–family conflicts—leading to fertility problems.

We did not detect significant differences in hematological o biochemical parameters. However, cholesterol tended to be lower in twin pregnancies that developed FGR. It has been demonstrated that birth weight is directly related to an elevation in lipids during pregnancy [19], and reduced placental transfer is associated with lower birth weight in FGR [20]. Therefore, in twin pregnancies, insufficient maternal lipid mobilization in early gestation may contribute to inadequate fetal growth. This hypothesis needs to be evaluated in a larger population.

During pregnancy, an increased oxygen demand elevates the rate of ROS production, and an adequate antioxidant balance is critical. This is particularly relevant around the 10th week of gestation, when the maternal circulation is established, and tissue oxygen requirements increase. Upon ROS elevation, there should be a parallel rise in antioxidants, and if they are insufficient, high oxidative damage might contribute to pregnancy disorders [9]. This is in agreement with our previous study demonstrating that a low antioxidant status in early pregnancy is associated with the development of maternal complications [12]. In the present study, we found that the antioxidant score (Antiox-S) was lower in women who developed FGR. This score may be useful to assess the antioxidant status of a subject, reducing the impact of biological variations in individual parameters, as we have previously demonstrated in populations with different pathologies, such as venous insufficiency [21], hypertension [22], or pregnancy complications [12]. The present findings suggest that, if maternal antioxidants are insufficient in twin pregnancies, oxidative damage might compromise fetal growth. In favor of this hypothesis are the evidence of increased lipid peroxidation in FGR pregnancies at the end of gestation [23] and the presence of higher F2-isoprostanes in the amniotic fluid from FGR pregnancies by the 18th week of gestation [24]. In our study, we did not find differences in individual oxidative damage biomarkers or in the integrated score Prooxy-S. We suggest that this may be related to the fact that we evaluated these parameters in early pregnancy, when it is unlikely that oxidative processes are already elevated. Therefore, it would be interesting to conduct a study to analyze the evolution of Antiox-S and Prooxy-S during the gestational period as well as their relationship with fetal outcomes.

Melatonin is an efficient antioxidant hormone that diffuses through biological membranes, exerting pleiotropic actions. Since oxidative stress is involved in pregnancy complications, melatonin is being considered as a treatment to reduce placental oxidative stress in preeclampsia and FGR [25]. Our data showed that lower melatonin levels were associated with preterm labor in twin pregnancies. These results are in agreement with reports of increased risk of preterm labor in women with shift work, which disrupts melatonin secretion [15]. We did not detect differences in melatonin levels between women with FGR and those with normal fetal growth. This is in contrast with a recent study which found a low concentration of melatonin in the third trimester of FGR pregnancies [26]. It is possible that this may be related to differences in the periods of gestation studied, since melatonin concentrations in humans are lower in the first trimester, with a rise after 24 weeks, reaching a maximum by the end of pregnancy [27]. We also found that maternal melatonin associated positively with birth weight. Our results are in accordance with several studies, which have highlighted the vital role of this hormone to preserve normal fetal growth and organ development [15,27,28].

Maternal cortisol showed the opposite trend as melatonin, i.e., an inverse association with birth weight. These data are in accordance with a recent meta-analysis [29]. At the 10th week of gestation, the fetus does not produce cortisol by itself [30], and therefore, cortisol’s effects on fetal growth depend on the maternal levels of this hormone. Elevation in cortisol levels is associated with psychosocial stress. It has been described that the use of ART is perceived by the mother as a stressful situation [31], and we have previous evidence that stress perception is greater in women carrying a twin pregnancy or an ART-derived pregnancy (unpublished observations). In physiological conditions, the fetus is protected from an excess of maternal cortisol through the placental barrier enzyme 11-β-HSD2. However, under situations of stress, this enzymatic system has been shown to be reduced [17], and its inactivation has been associated with FGR [18]. We did not have the possibility to evaluate alterations in placental–11-β-HSD2, and the analysis of the relationship between maternal stress and inactivation of the placental barrier deserves further study. One possibility could be the disruption of circadian rhythms by maternal stress, which could alter both cortisol and melatonin levels [27]. Our findings regarding the relationship between these hormones and birth weight suggest that it would be appropriate to monitor cortisol and melatonin levels in early pregnancy. Furthermore, it would be also interesting to evaluate psychological aspects in these women, mainly in those undergoing ART, who have a higher risk of stress.

### Strengths and Limitations of the Study

A strength of this work is the focus on twin pregnancies, since specific research is lacking. Another important point is the methodology design; the use of plasma samples in the first trimester, prior to the development of any obstetric problem, adds strength to the predictive potential of the molecules under evaluation. Thirdly, we confirm our previous findings in other populations of the usefulness of the antioxidant score as an approach to assess the oxidative status in plasma.

Limitations of the study are the homogeneity of the population and the analysis of participants from a single center. It would be desirable to confirm the present data in a larger population and in pregnant women with different characteristics (ethnicity, age), in order to be able to conclude on the predictive capacity of the antioxidant score. Similarly, it would be interesting to carry out a longitudinal study with different determinations throughout the pregnancy.

## 5. Conclusions

In conclusion, in twin gestations, low maternal plasma global antioxidant status and melatonin levels in early pregnancy could be considered potential biomarkers to predict FGR and preterm labor. Melatonin and cortisol could be potentially used to predict birth weight.

## Figures and Tables

**Figure 1 antioxidants-09-00269-f001:**
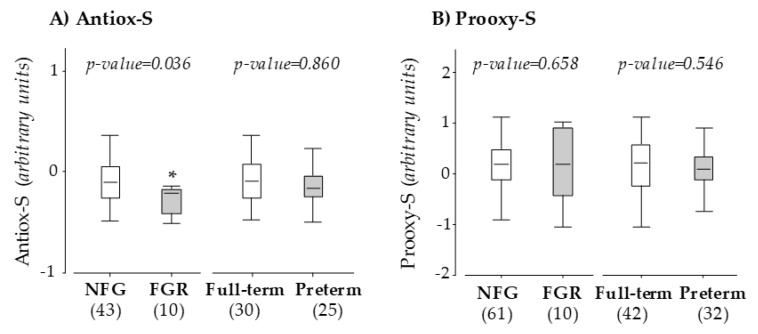
Maternal Antiox-S (**A**) and Prooxy-S (**B**) at 10 weeks of gestation. Data compare pregnant women with normal fetal growth (NFG) versus fetal growth restriction (FGR) and women with full-term versus preterm labor. Graphs show the median and interquartile range (IQR). Sample size is shown between brackets; **p* < 0.05, U Mann–Whitney test.

**Figure 2 antioxidants-09-00269-f002:**
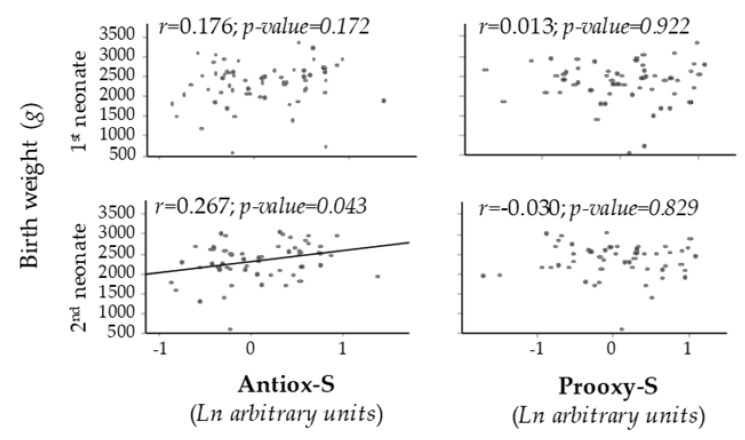
Scatter plots of birth weight vs. maternal Antiox-S or Prooxy-S values. Line shows *p* < 0.05. Rho–Spearman test.

**Figure 3 antioxidants-09-00269-f003:**
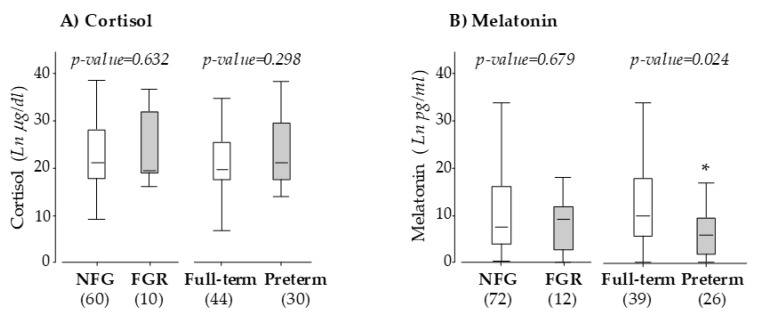
Maternal plasma cortisol (**A**) and melatonin (**B**) at 10 weeks of gestation. Data compare pregnant women with normal fetal growth (NFG) versus fetal growth restriction (FGR) and women with full-term versus preterm labor. Graphs show the median and IQR. Sample size is shown between brackets; **p* < 0.05, U Mann–Whitney test.

**Figure 4 antioxidants-09-00269-f004:**
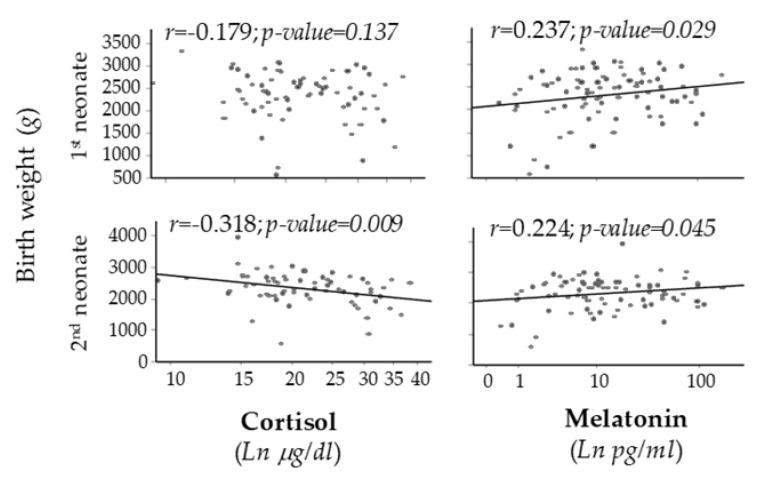
Scatter plots of birth weight vs. cortisol or melatonin levels. Line shows *p* < 0.05. Rho–Spearman test.

**Table 1 antioxidants-09-00269-t001:** Maternal and neonatal characteristics of the cohort.

Maternal Age *(years)*	35.1 ± 0.5
ART *(%)*	66.7 (68)
FGR *(%)*	13.6 (14)
Birth weight 1st neonate (*g*)	2339.8 ± 56.3
Birth weight 2nd neonate (*g*)	2323.9 ± 55.7
Gestational age (*weeks*)	36.1 ± 0.2
Preterm labor *(%)*	41.7 (43)

Data show mean ± SEM or relative frequency (sample size). ART, assisted reproduction techniques; FGR, fetal growth restriction.

**Table 2 antioxidants-09-00269-t002:** Maternal blood parameters, according to the development of FGR or preterm labor.

Parameter	NFG (60)	FGR (10)	*p*-Value	Full-Term (52)	Preterm (35)	*p*-Value
Glucose (mg/dL)	91.3 ± 3.4	96.9 ± 10.2	*0.554*	91.5 ± 4.3	95.7 ± 4.8	*0.520*
Cholesterol (mg/dL)	182.0 ± 3.8	169.0 ± 6.4	*0.183*	180.0 ± 4.0	182.0 ± 5.3	*0.725*
Triglycerides (mg/dL)	97.9 ± 3.2	95.5 ± 7.8	*0.580*	96.1 ± 3.6	100.2 ± 4.8	*0.412*
Hematocrit (%)	38.3 ± 0.4	39.9 ± 0.9	*0.170*	38.8 ± 0.5	38.1 ± 0.5	*0.310*
Leucocytes (10^6^/mL)	8.7 ± 0.2	8.7 ± 0.7	*0.967*	8.6 ± 0.2	8.9 ± 0.3	*0.538*

Data show mean ± SEM. Sample size shown between parentheses. NFG, normal fetal growth, FGR, fetal growth restriction. Student’s T test.

**Table 3 antioxidants-09-00269-t003:** Maternal plasma antioxidants and biomarkers of oxidative damage, according to the development of FGR or preterm labor.

Parameter	NFG (72)	FGR (12)	*p*-Value	Full-Term (52)	Preterm (39)	*p*-Value
Catalase activity (U CAT/mg prot.)	0.4 ± 0.03	0.5 ± 0.2	0.384	0.5 ± 0.05	0.4 ± 0.07	0.550
SOD activity (U SOD/mg prot.)	1.3 ± 0.1	1.1 ± 0.3	0.598	1.4 ± 0.2	1.2 ± 0.2	0.518
Glutathione (mg GSH/mg prot.)	0.7 ± 0.09	0.9 ± 0.2	0.440	0.8 ± 0.1	0.9 ± 0.1	0.407
Thiol groups (µM GSH/mg prot.)	4.7 ± 0.1	4.8 ± 0.6	0.684	4.7 ± 0.2	4.6 ± 0.2	0.778
Phenolic compounds (mg GAE/L)	300 ± 6.7	271 ± 26.4	0.149	290 ± 8.4	294 ± 10.5	0.805
Malondialdehyde (µmol/L)	6.5 ± 0.5	8.8 ± 1.7	0.101	7.2 ± 0.6	6.4 ± 0.6	0.359
Carbonyl groups (nmol/mg prot.)	0.9 ± 0.1	0.7 ± 0.2	0.530	0.8 ± 0.1	0.8 ± 0.1	0.626

Data show mean ± SEM. Sample size shown between brackets. CAT, catalase; SOD, superoxide dismutase; GSH, reduced glutathione; GAE, gallic acid equivalents; prot., protein. Student´s T test.

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
