# Peer review of "Maternal Antioxidant Status in Early Pregnancy and Development of Fetal Complications in Twin Pregnancies: A Pilot Study"

_antioxidants, 2020, doi:10.3390/antiox9040269_

Round 1

Reviewer 1 Report

This is an interesting and well conducted pilot study examining a range of prooxidant and antioxidant biomarkers in early gestation in twin pregnancies. Correlations are then drawn with the development of IUGR and pre-term birth.  Additionally, selected hormones were analysed in the sample set. By combining several biomarkers into an algorithm it was found that lower antioxidant status was associated with IUGR.  Lower melatonin levels were associated with pre-term birth.

Whilst these observations confirm other reports – one of the strengths of this study is the early and controlled collection of samples at week 9-11. This is a strong indicator that these changes maybe contributing to the development of these conditions. The other strength is the focus of twin pregnancies as most of the previous reports of oxidative changes during gestation have been on singleton pregnancy. The sample analysis and data analysis appears to be appropriate for this type of study. Overall, it makes a significant contribution, albeit a pilot study, and deserves publication.

I have no specific questions or problems with the study but I did notice several typos and a couple of grammatical errors which should be picked up during the editorial process.

Author Response

Thank you for your comments. We have revised the manuscript and tried to correct the grammatical errors and typos. They have been highlighted so they are easy to track.

We have also modified the scattered plots for clarity, eliminating lines from the graphs with no significant relationships.

Reviewer 2 Report

The authors presents a pioneering study on antioxydants in twin pregnancy.

The ms overall is well-written and well strcutured and mertis publication.

I would suggest to change the term IUGR as the one more recently introduced of FGR (fetal growth restriction).

The authors have also clearly reported potential bias and/or limitation of the current communication as well as its strength.

Author Response

Thank you for your comments. We have revised the manuscript and changed IUGR to FGR throughout the text and figures. The changes have been highlighted so they are easy to track.

We have also modified the scattered plots for clarity, eliminating lines from the graphs with no significant relationships.

Reviewer 3 Report

 Maternal antioxidant status in early pregnancy and 2 development of fetal complications in twin 3 pregnancies: a pilot study

This is a single-center, prospective, observational pilot study, including 104 twin healthy  pregnant women attended at the Obstetrics and Gynecology Service from La Paz University Hospital  (HULP, Madrid, Spain).

Well done study, several typos antioxidants should be anti-oxidants, in results not differences should be no differences, Data compare twin (not sure what should be said here).

Methods are appropriate and well described, figures are necessary

Only minor revisions needed

Author Response

Thank you for your comments. We have revised the manuscript, included the suggested changes and corrected other possible typos. The changes have been highlighted so they are easy to track.  

Line 180, we changed “not differences” to ”no differences”

Figure legend 1 and 3. The legends were confusing and we have deleted the word “twin”. Both legends now read: ”Data compare pregnant women with normal fetal growth (NFG) versus fetal growth restriction (FGR), and women with full-term versus preterm labor…..”.

We have also modified the scattered plots for clarity, eliminating lines from the graphs with no significant relationships.